# Early Nutritional Interventions with Zinc, Selenium and Vitamin D for Raising Anti-Viral Resistance Against Progressive COVID-19

**DOI:** 10.3390/nu12082358

**Published:** 2020-08-07

**Authors:** Jan Alexander, Alexey Tinkov, Tor A. Strand, Urban Alehagen, Anatoly Skalny, Jan Aaseth

**Affiliations:** 1Division of Infection Control and Environment Health, Norwegian Institute of Public Health, P.O. Box 222 Skøyen, 0213 Oslo, Norway; Jan.Alexander@fhi.no; 2Laboratory of Biotechnology and Bioelementology, Yaroslavl State University, Sovetskaya Str. 14, Yaroslavl 150000, Russia; tinkov.a.a@gmail.com (A.T.); skalny3@gmail.com (A.S.); 3IM Sechenov First Moscow State Medical University (Sechenov University), Bolshaya Pirogovskaya St., Moscow 119146, Russia; jaol-aas@online.no; 4Centre for International Health, University of Bergen, P.O. Box 7804, 5020 Bergen, Norway; tors@me.com; 5Research Department, Innlandet Hospital Trust, P.O. Box 104, 2381 Brumunddal, Norway; 6Division of Cardiovascular Medicine, Department of Medical and Health Sciences, Linköping University, SE-58185 Linköping, Sweden

**Keywords:** COVID-19, corona virus, nutritional, therapy, micronutrients, selenium, zinc, vitamin A, vitamin D, coenzyme Q_10_

## Abstract

Objectives: The novel coronavirus infection (COVID-19) conveys a serious threat globally to health and economy because of a lack of vaccines and specific treatments. A common factor for conditions that predispose for serious progress is a low-grade inflammation, e.g., as seen in metabolic syndrome, diabetes, and heart failure, to which micronutrient deficiencies may contribute. The aim of the present article was to explore the usefulness of early micronutrient intervention, with focus on zinc, selenium, and vitamin D, to relieve escalation of COVID-19. Methods: We conducted an online search for articles published in the period 2010–2020 on zinc, selenium, and vitamin D, and corona and related virus infections. Results: There were a few studies providing direct evidence on associations between zinc, selenium, and vitamin D, and COVID-19. Adequate supply of zinc, selenium, and vitamin D is essential for resistance to other viral infections, immune function, and reduced inflammation. Hence, it is suggested that nutrition intervention securing an adequate status might protect against the novel coronavirus SARS-CoV-2 (Severe Acute Respiratory Syndrome - coronavirus-2) and mitigate the course of COVID-19. Conclusion: We recommended initiation of adequate supplementation in high-risk areas and/or soon after the time of suspected infection with SARS-CoV-2. Subjects in high-risk groups should have high priority as regards this nutritive adjuvant therapy, which should be started prior to administration of specific and supportive medical measures.

## 1. Introduction 

The novel coronavirus SARS-CoV-2 (Severe Acute Respiratory Syndrome-coronavirus-2), causing COVID-19, is by far the most dangerous coronavirus ever identified, capable of infecting not only animals, but also humans across the globe. The severity of the COVID-19 pandemic has dramatically surpassed the prevalence of acute respiratory syndrome coronavirus (SARS-CoV) and Middle East respiratory syndrome coronavirus (MERS-CoV), which were distributed to more limited regions in 2003 and 2012, respectively. A single-stranded RNA comprises the genomic structure of SARS-CoV-2 [1]. In severe cases, COVID-19 is accompanied by excessive activation of the innate immune system with progressive inflammation and a cytokine storm from activated cells, particularly in the airways [2], leading to the cytokine release syndrome [3,4]. Unfortunately, in spite of their anti-inflammatory effects, corticosteroids have been observed to worsen the clinical status of patients with SARS or related virus infections [5,6]. Use of convalescent plasma has been tried as a possible approach, but the experiences with this strategy are limited [7]. Except for the use of convalescent plasma, there is at present no approved treatment or vaccine for COVID-19. Therefore, it is an urgent need for public health measures, not only to limit the spread of the virus, but also to implement preventive approaches to alleviate severe COVID-19, e.g., by reduction of the excessive inflammation. The metabolic status of the host, as influenced by advanced age, current medical condition, and lifestyle, appears to determine the clinical severity of COVID-19 [8]. In critically ill patients, coexisting diseases include type 2 diabetes, hypertension, and heart disease [9]. The elderly are more prone to severe respiratory infection than young people, apparently due to connections between old age and deficient nutrition and immunity [10]. Clinical and subclinical micronutrient deficiencies common in older adults are known to contribute to decreased immune function and age-related diseases [11], implying that nutritional management is essential to reduce the risk of severe infection [12]. In view of a lack of clinical data on preventive and/or therapeutic efficiency of the nutritive adequacy of selenium, zinc, and vitamin D in COVID-19, we, in the present narrative review, discussed recent clinical data on the role of these micronutrients in the protection against bronchopulmonary infections, as well as the existing indications of their impact on COVID-19. Although the status of other nutrients, such as vitamins C and A, may also play a role, they were not focused upon in the present article. We did a literature search for the period 2010–2020 on PubMed, Medline, and Google Scholar with the keywords of SARS, SARS-CoV-2, COVID 19, coronavirus, micronutrients (zinc, selenium, vitamin D), immune system, inflammation, prevention, and treatment. Based on the information retrieved, we here discussed the role of the nutritional status of certain trace elements and vitamin D in the perspective of principles for implementing preventive measures against RNA viruses. 

## 2. Nutritional Interventions as a Preventive Approach

Clinical or subclinical micronutrient deficiencies, such as deficiencies of zinc, selenium, and vitamin D, which frequently occur in old age groups, contribute to age-related diseases including diabetes, hypertension, and coronary heart disease [13,14,15]. These diseases, which in a substantial fraction of the cases are related to the metabolic syndrome [16], are characterized by signs of low-grade inflammation, which may also result from ageing [17]. Pre-infectious signs of inflammation, such as elevated values for CRP (C-reactive protein), represent a common aggravating factor in COVID-19 [9]. Adequacy of zinc, selenium, and vitamin D is essential for adequate immunocompetence, which to some extent may counteract an inflammatory aggravation. Dietary advice alone may not be sufficient to secure adequacy for these nutrients in certain conditions, including in elderly subjects [18], involving the need for supplements in susceptible segments of populations. 

### 2.1. Zinc

Being an essential component of numerous enzymes, such as superoxide dismutase 1 and 3 [19], the trace metal zinc is important for the development and maintenance of immune and other cells [20]. Zinc deficiency is known to result in dysfunctional humoral and cell-mediated immunity [21]. In the elderly, low Zn status (serum Zn values <0.7 mg/L) has been found to represent a risk factor for pneumonia [22]. Long-term zinc deficiency is known to increase inflammations and inflammatory biomarkers [23]. Most facets of the immune system are affected by zinc deficiency, particularly the T-cell function. Zinc deficiency also drives a Th17 response, which is associated with increased inflammation [24]. In elderly subjects, reduced concentrations of circulating zinc correlated with increased levels of the cytokines IL-6 (interleukine-6), IL-8, and TNF-α (Tumour necrosis factor-α) [25]. 

In a case report, four COVID-19 outpatients 26–63 years of age were treated with lozenges of zinc salts [26]. They took the lozenges several times each day, up to doses between 115 to 184 mg Zn/day for 10 to 14 days, and all of these patients recovered. In another case report, three COVID-19 patients 38–74 years of age with additional gut manifestations received zinc sulphate (220 mg Zn daily for 5 days), together with hydroxychloroquine and azithromycine [27]. The latter patients recovered. Being case reports, it was not possible to conclude on the efficacy of zinc. 

With regard to other infectious diseases, many studies show that zinc status may impact the outcome. Several randomized control trials (RCTs) showed that zinc given during an acute episode of diarrhea reduces the duration and risk of persistent disease [28]. The World Health Organization therefore changed their recommendations for the treatment of childhood diarrhea to include oral zinc medication. Zinc also plays a role in acute respiratory infections [29]. Routine zinc supplementation reduces the incidence of acute lower respiratory infections in young children in low- and middle-income countries [30]. Several recent studies used zinc as an adjunct treatment for lower respiratory infections, although with mixed results [31]. In one large RCT from India enrolling young infants with signs of severe bacterial illness, it was investigated whether zinc could reduce the risk of treatment failure [32]. The authors found that children assigned to the zinc group had a 40% reduction in treatment failure and mortality compared with the placebo group. Many RCTs examined the role of zinc supplementation in common colds, the results from these showing that, when given early in the illness, zinc had the potential to reduce the duration by 1 to 3 days [33,34,35]. Furthermore, a positive effect of zinc supplementation was observed in several studies on hepatitis C, which is induced by infection with a single-stranded RNA virus [36]. In this context, it is of interest that raising the intracellular concentration of zinc with zinc-ionophores like pyrithione or chloroquin could directly reduce the replication of a variety of RNA viruses in cells in vitro through inhibition of their RNA polymerase activity [37]. Combined administration of zinc and pyrithione, even at low concentrations, inhibited the replication of SARS coronavirus (SARS-CoV) in vitro [38]. Consequently, zinc supplement may have effects, not only on the COVID-19-associated over-active inflammation, but presumably also on the SARS-CoV-2 agent itself [39]. As for the preventive doses used, it was noted that, on a long-term basis, an intake ≤25 mg/day was recommended, as a high intake of zinc may disturb copper balance [40].

### 2.2. Selenium

Selenium is an essential trace element for mammalian redox biology by occurring as selenocysteine in catalytical centers of many selenoproteins [41,42]. An adequate supply of the amino acid serine is required for the synthesis of selenocysteine, which is incorporated into selenoproteins [43]. Nutritional deficiencies of selenium may impact, not only the immune response, but also the pathogenicity of a virus [44,45,46].

Of note, a recent study from China reported an association between the cure rate of CoV-2-infected patients and selenium status, as deduced from city population hair selenium from cities outside Hubei, reflecting regions with poor and adequate selenium intakes [44]. In a study, selenium status (selenium and SELENOP) were significantly higher in surviving COVID-19 inpatients (*n* = 27) compared with non-survivors (*n* = 6) [45]. Further studies with control of confounding and clinical trials are necessary to confirm this association. Of particular interest is the finding that a main protease of SARS-CoV-2 responsible for the viral replication, interacts with the essential seleno-enzyme glutathione peroxidase1 (GPX1) [46,47], which is strongly dependent on adequate selenium supply. It is notable that the GPX mimic ebselen (a synthetic selenium compound) is a potent inhibitor of the SARS-CoV-2 main protease [48]. Bioinformatic screening of the SARS-CoV-2 gene signatures provided further evidence of protein interactions and antisense transcript mRNA–mRNA interactions occurring at selenocysteine-related insertions in RNA viruses [49]. 

Dietary selenium deficiency, together with increased oxidative stress in the host, can alter a viral genome from a normally mildly pathogenic virus into a highly virulent agent after its entrance into the host, which occurred with the Coxsackie 3B virus in Keshan disease in a selenium-deficient area in China [50]. It was proposed that Se deficiency could play a substantial role in the genesis of SARS-CoV [51]. The potential protective effect of selenium is explained by its role as an essential cofactor in a group of enzymes that, in concert with vitamin E, works to reduce the formation of reactive oxygen species (ROS). ROS in excess may trigger oxidative changes both in invading microorganisms and in the cells in the host [52]. 

A failing antioxidant defense might also be accompanied by an exaggerated inflammatory response in the host, even in the absence of an active infection [53]. Among the most potent antioxidative selenoenzymes are the glutathione peroxidases (GPXs) and the thioredoxin reductases (TXNRDs), which need an intake of at least 100 µg Se/day to function optimally. 

Other selenoproteins, i.a., selenoprotein K (SELENOK) and selenoprotein S (SELENOS), also appear to play a role in the regulation of immune responses [54]. 

In a variety of infectious diseases selenium appears to play a significant role in protecting the respiratory system, in particular toward viral infections [54]. Beck et al. found that Se deficiency significantly increased the susceptibility to influenza-induced lung pathology associated with the overexpression of pro-inflammatory cytokines [55]. An analogous effect was observed in benign Coxsackie virus infection, which resulted in the development of myocarditis in Se-deficient mice [56]. These findings corresponded to the observation of lower interferon-γ (IFN-γ) and TNF-α levels, as well as reduced survival rate in Se-deficient mice infected with the influenza virus as compared to Se-adequate controls [57]. In turn, selenium treatment was shown to up-regulate the expression of genes for interferons (IFN-α, IFN-β, and IFN-γ) in response to the avian influenza (H9N2) virus [58]. 

In older adult humans, Se treatment was shown to modulate response to the influenza vaccination, being accompanied by increased IFN-γ levels after vaccination [59]. Therefore, selenium supplementation to populations with suboptimal status has been considered a safe adjuvant therapy in preventive measures against viral infections [60]. The selenium status varies widely between different areas in the world. Compared with levels in Northern America [61], selenium levels in populations in large parts of Europe are well below a threshold of about 100 µg/L required for adequate expression of selenoproteins. The insufficient selenium intake is caused by low selenium content in soil and, consequently, in cereals and other food plants, as well as in fodder for grazing farm animals [41,62].

### 2.3. Selenium Plus Cofactors

The optimal function of the GPXs also depends upon adequate intracellular levels of the cofactor glutathione (GSH), explaining the importance of adequate intakes of proteins containing the sulfur component of this tripeptide, viz. cysteine or methionine. Reduced GSH is associated with senescence in several species, including humans [63]. Apparently healthy elderly people in the age group 60–79 had significantly lower erythrocyte GSH than younger individuals [64]. Moreover, individuals with chronic diseases, including hypertension, have a deficit of the active form of GSH [65,66]. In cases of marginal intakes of sulfur amino acids, supplementation with acetylcysteine will restore intracellular GSH levels, which is of crucial importance in bronchial and pulmonary cells [67]. N-acetylcysteine is already an approved and extensively used drug in obstructive bronchitis [68], and it has proven beneficial against severe influenza infection [69]. Administration of glutathione has been shown to relieve dyspnea associated with COVID-19 pneumonia [70]. 

Another factor co-operating with selenoenzymes appears to be coenzyme Q_10_ (CoQ_10_). In a Swedish randomized placebo-controlled study, healthy elderly subjects low in selenium were given selenium supplementation combined with coenzyme Q_10_. This supplementation was shown to reduce the non-specific inflammatory response as measured by plasma CRP [71] and other biomarkers of inflammation [53], and also cardiovascular mortality [72]. As severe coronavirus infections are characterized by an overactive inflammation, this relief in inflammatory response by optimizing the selenium status is of considerable interest [62]. It is also relevant that CoQ_10_ supplements, even when given alone, can exert an anti-inflammatory effect [73]. An anti-inflammatory effect of exogenous CoQ_10_ may appear clearer in old age when its endogenous production is significantly reduced [74]. 

Selenium treatment given alone, without combination with acetylcysteine or CoQ_10_, in critical ill patients admitted to the intensive care unit (non-septic and septic patients) has been used [75]. In patients with advanced infections, Manzanares and coworkers [76], in a meta-analysis, did not find a consistent beneficial effect on mortality, but, in a subgroup analysis, they found a reduction in the infections in non-septic patients. When considering the positive effects of selenium on immune regulation and inflammation in populations low in selenium, it appears justified to conclude that an adequate pre-infectious status of selenium would represent a protective measure against the hyperinflammation characterizing corona viral infections. Thus, in subjects with suboptimal status (plasma selenium <100 µg/L), supplementation at a dose of 100–200 µg Se/day, with or without cofactors, to achieve rapid saturation of vital selenoproteins, should represent an adjuvant approach to prevent aggressive SARS-CoV-2 infection. However, a total long-term intake of selenium from food and supplements ≤300 µg Se/day is recommended, as higher intakes may be associated with toxicity [77]. 

### 2.4. Vitamin D

It is well-known that cholecalciferol (vitamin D3) can be synthesized from cholesterol in the body skin upon exposure to sunlight. Its biological activity is dependent on successive hydroxylations by the liver and the kidneys to 1,2-(OH)_2_-D3, which binds to vitamin D receptors. Beyond its roles in calcium homeostasis and the maintenance of bone integrity, it also stimulates the maturation of immune cells. Epidemiological studies suggested an inverse association between circulating levels of 25(OH)-D3, a biomarker of vitamin D status, and inflammatory biomarkers, including CRP and IL-6 [78]. Suboptimal levels of vitamin D, particularly at the end of the winter season, have been reported in a substantial number of otherwise healthy adults [79]. People with limited access to sunlight, and elderly with reduced synthesizing capacity, may have vitamin D deficiency [80]. 

Vitamin D has been suggested to play a role in COVID-19, as two ecological studies indicated that the rate of infection was higher in countries at higher latitudes and/or lower vitamin D status [81,82]. In a non-peer-reviewed study from Los Angeles, vitamin D deficiency was identified as a risk factor for positive COVID-19 tests [83]. A recent study on COVID-19 inpatients (*n* = 134) found that a significantly smaller fraction of patients in intensive care units had 25-OH-D above 50 nmol/L (19%) compared with those in conventional medical wards (39.1%) [84]. In a non-peer-reviewed study from Cincinnati, the authors found associations between vitamin D deficiency and hospital admission, disease severity, and also with death, among patients from primary care and specialized clinics (*n* = 691) [85].

Vitamin D was shown to be an essential factor for protection against respiratory infectious diseases [86]. Severe vitamin D deficiency is frequently seen in critically ill patients and appears to be related to poor prognosis [87]. In older patients, severe vitamin D deficiency has been considered an independent predictor for community-acquired pneumonia [88], being also associated with increased risk of admission to an intensive care unit [89], and associated with mortality [90]. Moreover, vitamin D deficiency is shown to be associated with aggravation of lung inflammation, leading to acute respiratory distress syndrome (ARDS) with respiratory epithelium damage and hypoxia [91]. An inverse association between 25-OH-cholecalciferol levels and risk of acute respiratory failure in critically ill patients has been observed, being most convincingly significant for subjects with 25-OH-cholecalciferol <25 nmol//L [92]. 

Increasing experimental data on cells in vitro demonstrated beneficial effects of vitamin D as to pathogenetic mechanisms of respiratory viral infections. Thus, vitamin D treatment was shown to reduce respiratory syncytial virus (RSV) and rhinovirus (RV) replication in epithelial cells through enhancement of virus-induced interferon-stimulated genes [93] and synthesis of the antiviral protein LL-37 [94]. Treatment with 1,25(OH)-D improved respiratory-induced antiviral immune response to RV infections characterized by up-regulation of IL-8 and CXCL-10 (C-X-C motif chemokine ligand 10 also known as Interferon gamma-induced protein 10) production [95]. 

In addition, it was demonstrated that vitamin D is capable of reducing inflammatory response without alteration of antiviral activity and viral clearance in airway epithelial cells infected with RSV [96]. Furthermore, in view of high incidence of lung fibrosis as a characteristic sequela of COVID-19 [97], it is important to note that vitamin D prevented a TGF-β1-induced profibrotic phenotype of lung cells [98].

However, no preventive effect of vitamin D supplementation on pneumonia was observed in three independent case-control studies [99], but the interpretation of these results should take into account the apparent lack of pre-existing vitamin D deficiency. Furthermore, the benefit of vitamin D replacement in an advanced stage of critical illness is controversial, as some studies do not show a benefit when it is administered late in the critical disease [100]. 

Vitamin D status can easily be determined as 25-OH-cholecalciferol in plasma. It follows that, in case of low status, <50 nmol/L in plasma, vitamin D supplementation (40 µg D_3_/day) could work as an approach for prevention of an aggressive course of the inflammation induced by this novel coronavirus. As for the preventive doses used, it is recommended that, on a long-term basis, the intake of vitamin D should be ≤100 µg D3/day to avoid hypercalcuria with risk of renal stones, and also hypercalcemia [101].

## 3. Discussion and Conclusions

The direct evidence that the micronutrients zinc, selenium, and vitamin D might be involved in the course and outcome of the COVID-19 disease is observational and weak. However, based on experiences from treatments of SARS and other viral infections, we here underscored observations showing that nutritive supplements administered at an early stage of the infection were important for enhancing host resistance against RNA viral infections, which might also include severe COVID-19. We hypothesized that, in particular, increased resistance toward escalation of COVID-19 into the life-threatening cytokine release syndrome might be obtained (Figure 1). The nutritional status of the host has yet not been considered a crucial factor in severe viral infections, because the efficacy of nutrient supplementation when administered at the stage of advanced illness has been disappointing. Nevertheless, it is conceivable that a good nutritional status, if achieved in vulnerable population segments before escalation of the disease, would have immuno-enhancing and anti-inflammatory effects [102]. We are aware of the alleged therapeutic role of megadoses of vitamin C (6–8 g/day) in viral infections [103,104], but, as this would be a pharmacological approach, we did not further discuss this in the present article. We considered the proposed intervention with, i.a., proteins and multivitamin solutions, given immediately after hospital admission to relieve the COVID-19 infection [105] to represent an interesting modification of our approach. However, further research and clinical trials are requested both on therapeutic and preventive roles of nutritive supplements. Based on the available literature, a reasonable presumption is that the pre-infectious status of zinc, selenium, and vitamin D might be of especial importance for the resistance against a progressive course of COVID-19. 

Our recommendations are early outpatient nutritional intervention in SARS-CoV-2 exposed or high-risk subjects, preferably before specific and supportive treatment. It is tempting to suggest that that early nutritional interventions will be of particular significance for vulnerable segments of populations in developing countries. Such an approach is simple, cheap, and harmless. While high doses of the micronutrients might be needed to restore deficiencies, it is advisable to follow recommended upper tolerable intake levels for long-term intakes of the micronutrients. Parallel to any of the nutritional approaches, controlled studies on the efficacy of anti-viral and anti-inflammatory measures are of importance. To obtain general immunity, a COVID-19-related vaccine is highly warranted. 

## Figures and Tables

**Figure 1 nutrients-12-02358-f001:**
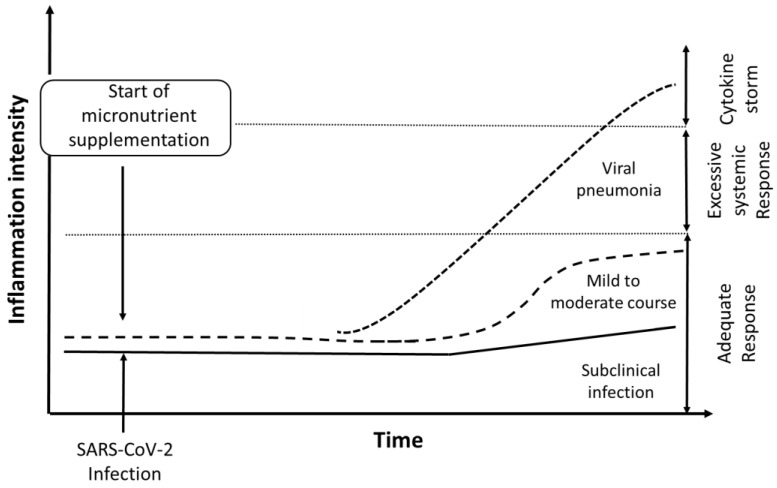
Hypothesized effect of supplements (Zn, Se, and vitamin D) on intensity of inflammation in patients with COVID-19: A severe course of the disease, which may occur in cases with pre-infectious low-grade inflammation and inadequate status of micronutrients, is characterized by an escalation of the inflammation into a cytokine storm (dotted line). Supplementation with Se, Zn, and vitamins when given at an early stage after infection is anticipated to act protectively by improving immune reactivity and supporting adequate inflammatory response, leading to lower risk of cytokine storm and less severe course of COVID-19, as indicated by the dashed line.

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
