# Peer review of "Early Nutritional Interventions with Zinc, Selenium and Vitamin D for Raising Anti-Viral Resistance Against Progressive COVID-19"

_nutrients, 2020, doi:10.3390/nu12082358_

Round 1
Reviewer 1 Report
The article; “Early nutritional interventions with zinc, selenium 3 and vitamin D for raising anti-viral resistance against 4 progressive COVID-19” by Alexander et al., is a timely overview of the literature regarding nutritional effects on resistance to viral diseases. The writing is mostly well done except for the following minor issues:
Minor comments:
Wording adjustments are needed for smooth reading, e.g.:
Line 57, change “represents” to “comprises” or similar modification
Line 63, change “presents” to “present”
Lines 77, 90, 96, change “i.a.” to “i.e.”
Similar minor grammatical and wording occur throughout the manuscript. At their discretion, the authors may wish to employ an English language editor to more conveniently eliminate these issues prior to resubmission.
Comments:
Line 107-109. I believe you intend to say more than simply that the patients recovered. Were there significant treatment dependent effects in comparison to the control group? Perhaps the treatment group recovered more quickly? If not, why report this? Don’t you intend to describe the effects of treatment on their recoveries?
Line 137, It may be good to include the citations (44-46) supporting the statement just made. You will be detailing the findings of the individual studies, but the statement concluded here needs citations.
Line 200-201, indicate the condition that was causing the critical illness in the patients that were successfully treated with supplemental Se. (I believe these were septicemia patients?)
Line 276-277. Perhaps you should clarify why you are choosing not to comment on the efficacy of Vitamin C megadoses?
Due to the fast-moving publications in this area, a couple more citations should be incorporated in the revised article:
Seale LA, Torres DJ, Berry MJ, Pitts MW. A role for selenium-dependent GPX1 in SARS-CoV-2 virulence. Am J Clin Nutr. 2020 Jun 27:nqaa177. doi: 10.1093/ajcn/nqaa177.
Moghaddam A, Heller RA, Sun Q, Seelig J, Cherkezov A, Seibert L, Hackler J, Seemann P, Diegmann J, Pilz M, Bachmann M, Minich WB, Schomburg L. Selenium Deficiency Is Associated with Mortality Risk from COVID-19. Nutrients. 2020 Jul 16;12(7):E2098. doi: 10.3390/nu12072098.
George D Vavougios. Selenium - associated gene signatures within the SARS-CoV-2 - host genomic interaction interface. Free Radic Biol Med. 2020 Jul 14;S0891-5849(20)31150-3.
Author Response
Reviewer 1
The article; “Early nutritional interventions with zinc, selenium 3 and vitamin D for raising anti-viral resistance against 4 progressive COVID-19” by Alexander et al., is a timely overview of the literature regarding nutritional effects on resistance to viral diseases. The writing is mostly well done except for the following minor issues:
Reply: Thank you
Minor comments:
Wording adjustments are needed for smooth reading, e.g.:
Line 57, change “represents” to “comprises” or similar modification
Reply: Line 57 Changed as suggested
Line 63, change “presents” to “present”
Reply: Line 63, changed as suggested
Lines 77, 90, 96, change “i.a.” to “i.e.”
Reply: The meaning of i.a. is “inter alia”. The wording has been changed avoiding this term, lines 77, 90, 102.
Similar minor grammatical and wording occur throughout the manuscript. At their discretion, the authors may wish to employ an English language editor to more conveniently eliminate these issues prior to resubmission.
Reply: We have gone through the manuscript and corrected minor language errors. Lines 63, 142, commas added
Comments:
Line 107-109. I believe you intend to say more than simply that the patients recovered. Were there significant treatment dependent effects in comparison to the control group? Perhaps the treatment group recovered more quickly? If not, why report this? Don’t you intend to describe the effects of treatment on their recoveries?
Reply: Line 116. These were case reports and conclusions as to efficacy cannot be made. This has been noted
Line 137, It may be good to include the citations (44-46) supporting the statement just made. You will be detailing the findings of the individual studies, but the statement concluded here needs citations.
Reply: Line 144: References inserted
Line 200-201, indicate the condition that was causing the critical illness in the patients that were successfully treated with supplemental Se. (I believe these were septicemia patients?)
Reply: Line 215. We have added information on patients studied
Line 276-277. Perhaps you should clarify why you are choosing not to comment on the efficacy of Vitamin C megadoses?
Reply: Line 291-292We have focused on nutritional intervention. Megadoses are pharmacological doses. This has been noted
Due to the fast-moving publications in this area, a couple more citations should be incorporated in the revised article:
Seale LA, Torres DJ, Berry MJ, Pitts MW. A role for selenium-dependent GPX1 in SARS-CoV-2 virulence. Am J Clin Nutr. 2020 Jun 27:nqaa177. doi: 10.1093/ajcn/nqaa177.
Reply: Line 155 Reference included
Moghaddam A, Heller RA, Sun Q, Seelig J, Cherkezov A, Seibert L, Hackler J, Seemann P, Diegmann J, Pilz M, Bachmann M, Minich WB, Schomburg L. Selenium Deficiency Is Associated with Mortality Risk from COVID-19. Nutrients. 2020 Jul 16;12(7):E2098. doi: 10.3390/nu12072098.
Reply: Reference 45 already included as a preprint – citation updated
George D Vavougios. Selenium - associated gene signatures within the SARS-CoV-2 - host genomic interaction interface. Free Radic Biol Med. 2020 Jul 14;S0891-5849(20)31150-3.
Reply: Lines 156-159 Reference included – with additional text and a new reference:
Sies, H.; Parnham, M.J. Potential therapeutic use of ebselen for covid-19 and other respiratory viral infections. Free Radic. Biol. Med. 2020, 156, 107–112
Reviewer 2 Report
This narrative review is well written and adds some insights to the issue of vitamin and micornutrient supplementation in COVID-19.
I would simply recommend the authors to be more prudent when referring to Figure 1, as what they propose could be just hypothesized at the moment.
I would change the text in Figure 1 accordingly and I would add a few statements in the discussion to underline the current limitations.
Author Response
Reviewer 2
This narrative review is well written and adds some insights to the issue of vitamin and micornutrient supplementation in COVID-19.
Thank you
I would simply recommend the authors to be more prudent when referring to Figure 1, as what they propose could be just hypothesized at the moment.
Reply: Line 284 Agree; it is referred to as hypothetical
I would change the text in Figure 1 accordingly and I would add a few statements in the discussion to underline the current limitations.
Reply: Line 313 Figure legend amended